# Genome-Wide Identification and Characterization of Lignin Synthesis Genes in Maize

**DOI:** 10.3390/ijms25126710

**Published:** 2024-06-18

**Authors:** Shuai Wang, Xiaofang Wang, Liangxu Yue, Huangai Li, Lei Zhu, Zhenying Dong, Yan Long

**Affiliations:** Zhongzhi International Institute of Agricultural Biosciences, Research Institute of Biology and Agriculture, School of Chemistry and Biological Engineering, University of Science and Technology Beijing, Beijing 100083, China; b20190394@xs.ustb.edu.cn (S.W.); b20200410@xs.ustb.edu.cn (X.W.); m202310926@xs.ustb.edu.cn (L.Y.); huangaili@ustb.edu.cn (H.L.); zhulei@ustb.edu.cn (L.Z.)

**Keywords:** maize, lignin, synthesis gene, transcription factor, lodging

## Abstract

Lignin is a crucial substance in the formation of the secondary cell wall in plants. It is widely distributed in various plant tissues and plays a significant role in various biological processes. However, the number of copies, characteristics, and expression patterns of genes involved in lignin biosynthesis in maize are not fully understood. In this study, bioinformatic analysis and gene expression analysis were used to discover the lignin synthetic genes, and two representative maize inbred lines were used for stem strength phenotypic analysis and gene identification. Finally, 10 gene families harboring 117 related genes involved in the lignin synthesis pathway were retrieved in the maize genome. These genes have a high number of copies and are typically clustered on chromosomes. By examining the lignin content of stems and the expression patterns of stem-specific genes in two representative maize inbred lines, we identified three potential stem lodging resistance genes and their interactions with transcription factors. This study provides a foundation for further research on the regulation of lignin biosynthesis and maize lodging resistance genes.

## 1. Introduction

Lignin is a class of polymeric aromatic compounds and the second most abundant natural organic substance after cellulose on Earth [1]. It is an important component of plant cell walls. Lignins are primarily divided into p-hydroxyphenyl lignin (H-lignin), guaiacyl lignin (G-lignin), and syringyl lignin (S-lignin), which are respectively polymerized from p-coumaryl alcohol, coniferyl alcohol, and sinapyl alcohol. They are interconnected via unstable ether bonds (β-O-4 bonds) and/or resistant carbon–carbon bonds (called condensed bonds) [2,3]. This closely cross-linked polymer structure plays an important role in maintaining the shape of plant cell walls, enhancing the mechanical strength of plant cell walls, and increasing plant lodging resistance [4,5,6,7,8,9]. For example, in rice, the cell wall components like cellulose, hemicellulose, and lignin contents are generally higher in an indica variety Huazhan than that in a japonica variety R2, and *Huazhan* has higher mechanical strength and stronger lodging resistance compared to R2 [10]. In wheat, *Qingmai* 1175 has stronger lodging resistance than that in Qingmai 18 [11]. In maize, the lignin synthesis-related genes and enzyme activities are higher during stem development, accelerating lignin accumulation and enhancing stalk lodging resistance [12]. Prolonging the period of rapid lignin accumulation can also significantly increase corn stalk strength and lodging resistance [13]. In addition to being a major component of cell walls, lignin has been found to be involved in various biological processes, such as plant growth and development [14,15], plant insect pest and disease resistance [16,17], plant heavy metal tolerance [18,19], plant drought, salt, and temperature stress tolerance [20,21]. Meanwhile, lignin also prevents microbial degradation of plant cell walls, and has played an important role in the evolution of plants [22,23,24].

Lignin is a product of the phenylpropanoid metabolic pathway, and its synthetic pathway can be divided into three stages [3,25,26,27]. In the first stage, plants assimilate glucose into aromatic amino acids like phenylalanine, tryptophan, and tyrosine through photosynthesis, which is called the shikimate pathway. In the second stage, phenylalanine is converted into corresponding products under the action of enzymes like phenylalanine ammonia lyase (PAL), cinnamate 4-hydroxylase (C4H), Coumarate 3-hydroxylase (C3H), Shikimate Hydroxycinnamoyl Transferase (HCT), caffeic acid O-methyltransferase (COMT), caffeoyl CoA O-methyltransferase (CCoAOMT), and ferulate 5-hydroxylase (F5H), and finally into corresponding coenzyme A thioesters by 4-coumarate CoA ligase (4CL). In the third stage, cinnamoyl CoA reductase (CCR) and cinnamyl alcohol dehydrogenase (CAD) convert the products from the second stage into three lignin monomers (Appendix A).

Lignin synthetic-related genes have been extensively studied in various crops [28,29,30,31]. The brown midrib phenotype in maize is associated with impaired lignin synthesis. The 4-coumarate CoA ligase gene *Zm4CL1* encoded by *bm5* directly affects G lignin synthesis but does not significantly affect maize growth and development [32,33]. *bm1* encodes *ZmCAD*. The characteristics of the bm1 mutant are relatively simple. In lignified tissues, its CAD activity is severely reduced, leading to changes in the total amount of lignin and the structure of lignin monomers [34]. In rice, the functional loss of two 4CLs, *Os4CL3* and *Os4CL4*, altered the composition of lignin polymer units differently. The loss of function of *Os4CL3* leads to a significant reduction in G-lignin and H-lignin units, with G-lignin being more significantly reduced than H-lignin. On the contrary, *Os4CL4* lacks the abundance of G-lignin units in its function. In addition, the loss of function of *Os4CL3* and *Os4CL4* significantly reduced the binding of ferulic acid salts to the cell wall, indicating their role in cell wall ferulicylation [35]. *OsAAE3* is a homolog of Arabidopsis AAE3 in rice, encoding the 4CL-like protein. Overexpression of *OsAAE3* inhibits the development of small flowers, exhibiting significant glume distortion and reduced anther fertility. Meanwhile, this leads to a decrease in lignin content [29]. In wheat, analysis of differentially expressed genes (DEGs) related to the phenylpropanoid biosynthesis pathway showed that lignin synthesis-related genes *4CL*, *CCR*, *CAD*, *PAL*, and *HCT* were significantly up-regulated in Qingmai 1175 [36]. CCR is the entry point of the lignin-specific branch of the phenylpropanoid pathway. Overexpression of lignin synthesis-related genes BnCCR1 and BnCCR2 in rapeseed led to significantly enhanced lodging resistance in transgenic rapeseed compared to wild type [11].

In addition to the lignin biosynthesis genes, transcription factors (TFs) like MYB and NAC are also involved in regulating lignin biosynthesis [26,37]. For example, in Brassica napus, BnMYB52a can specifically bind to the promoters of 4CL and CCR to regulate lignin synthesis [38]. The *OsNAC7* subfamily controls lignification in chrysanthemums, and its member *ClSND1* can thicken tobacco cell walls and increase secondary xylem thickness when overexpressed in tobacco [39]. In Arabidopsis, overexpression of two TFs, *MYB58* and *MYB63*, activated the expression of lignin biosynthesis-related genes and promoted the lignification of cells [40].

Maize is an important cereal crop that serves as a source of food, animal feed, and biofuel for humans [41,42]. Lignin plays an important role in maize production. However, the exact number of key enzyme genes involved in the lignin biosynthesis pathway in the maize genome remains unclear, hindering in-depth research on the mechanism of lignin biosynthesis in maize. In order to comprehensively identify lignin biosynthesis pathway genes and transcriptional regulators in the maize genome, we used Arabidopsis lignin biosynthesis pathway genes as a reference and successfully identified 10 gene families involved in lignin biosynthesis through homologous alignment. The gene structures, evolutionary relationships, and expression patterns of these genes were thoroughly analyzed, and potential TFs that may bind to the promoters of key enzyme genes were predicted. To further investigate lignin biosynthesis pathway genes and corresponding TFs involved in lodging resistance, two representative maize inbred lines with significant differences in stem strength were selected as materials. The potential candidate lignin biosynthesis genes were predicted by gene expression analysis in the stems of the two inbred lines. The results of this study not only shed light on maize lignin biosynthesis pathway genes at the genomic level but also identified key genes, providing a foundation for further research on the molecular mechanisms of lignin biosynthesis and regulation in maize. Additionally, this study offers valuable insights for studying lignin synthesis and lodging resistance in other plant species.

## 2. Results

### 2.1. Identification of Gene Families Involved in the Lignin Synthetic Pathway in Maize

We identified 117 synthetic genes of 10 gene families in the maize genome (Figure 1, Appendix A). Among these genes, 52 were annotated as known synthase genes, so we denoted the remaining 90 identified genes with the prefix “*Zm*” based on their specific position on the chromosomes (Chr.). After comparing the numbers of these gene family members between maize and *Arabidopsis*, we found that the number of genes in *Arabidopsis* was low (a total of 29 genes and only three members per gene family on average) (Appendix A). By contrast, the number of genes related to lignin synthesis in maize was significantly higher, with an average of 14 members per gene family (Appendix A). The gene member numbers of different gene families ranged from 2 to 35; the *CCR* gene family had the greatest number at 25 and the *COMT* gene family had the least number at 3.

To further elucidate the evolutionary relationships between each maize gene family member and their homologs in *Arabidopsis*, we constructed phylogenetic trees (Appendix A). The maize lignin synthesis gene family members with different numbers are on the same evolutionary branch as their corresponding *Arabidopsis* gene members, indicating a higher homology between lignin synthesis genes in maize and *Arabidopsis*.

Next, we annotated the specific position of each gene in the chromosome. We found that the 117 genes were distributed on 10 chromosomes, and they exhibited an uneven distribution pattern ranging from 5 to 17 genes per chromosome (Figure 1). Furthermore, the genes displayed significant clustered distribution patterns on the chromosomes. The *ZmPAL* gene families had the most clusters, while the *ZmCOMT* gene family had the least. Notably, the *ZmPAL3*-*ZmPAL5* of the *PAL* family were found to be repeatedly distributed (Figure 1).

### 2.2. Gene Expression Pattern Analysis of the Identified Synthetic Genes

To obtain a whole expression pattern picture of these identified synthetic genes, we analyzed the gene expression patterns by using RNA-Seq data from 21 different tissues (Appendix A). Based on their expression patterns, we divided the genes into five groups, including those mainly expressed in roots, stems, leaves, seeds, and flowers, respectively (Figure 2). Among the five groups, 48 genes were highly expressed in roots, 14 in stems, 14 in leaves, 21 in seeds, and 17 in flower. The results showed that the core genes in the phenylpropanoid pathway were highly expressed in most stem and root tissues, as these tissues have higher degrees of lignification.

### 2.3. Prediction of Transcriptional Regulatory Network in Lignin Synthetic Pathway

In order to highly predict the interactive TFs with synthesis genes, the method in PlantTF DB and the TGMI method were combined for analysis. Finally, we identified 151 TFs predicted to interact with 96 synthetic genes. These TFs could be divided into 32 types, including MYB, NAC, bHLH, WRKY, and so on. The type with the highest number is the WRKY TF, with 70; the following type is the NAC and MYB TFs, with the numbers 42 and 40, respectively (Appendix A). In previous studies, MYB and NAC are two types of TFs that have been found to be involved in lignin synthesis. So, we selected these two types of TFs for further analysis. Based on the interaction frequency between TF and the synthesis gene, 29 TFs that might potentially regulate 43 synthetic genes were identified (Figure 3A).

The MYB regulatory network comprised 40 nodes, including 15 TFs and 25 structural genes. The central TF *Zm00001eb248590*, *Zm00001eb014280*, and *Zm00001eb014430* interacted with the largest number of downstream genes at up to 5, while peripheral TFs such as *Zm00001eb041450*, *Zm00001eb103730*, and *Zm00001eb288730* linked to only one downstream gene, respectively, indicating a narrower regulatory scope. Within the NAC regulatory network, there were 82 nodes in total, comprising 14 TFs and 43 structural genes. The central TF Zm00001eb157260 connected with the most numbers of downstream genes (7 genes), while peripheral TFs such as *Zm00001eb213710*, *Zm00001eb024300*, and *Zm00001eb324550* were each linked to only one downstream gene, indicating minimal regulatory involvement (Figure 3B). Therefore, in the lignin synthesis process, different synthesis genes may interact with the same TFs.

### 2.4. Gene Expression Analysis of Lignin Synthesis Genes and TFs in Different Tissues

In order to identify highly expressed synthetic genes and their corresponding TF in lignin synthesis process, we selected four TF-synthetic gene combinations with the highest expression levels in stems, including *ZmTCPTF16/Zm00001eb098190*, *ZmWRKY98/ZmUMC2381*, *ZmBHLH88/Zm00001eb234090*, and *ZmMYB19/Zm00001eb308860* to perform RT-qPCR analysis in different developmental stages of B73 and Mo17 (Figure 4A,B). The results showed that the genes expressed with low levels in the stem during the sixth leaf stage in both of the two inbred lines (Appendix A). Subsequently, we collected tissues from B73 during the 14th leaf stage and found that, except for *ZmWRKY98*, the other genes had the highest expression levels in the stem (Figure 4C). This suggests that the expression patterns of these genes increase after stem morphogenesis.

### 2.5. Candidate Lignin Synthesis Gene Analysis

To further investigate the role of these potential lignin synthesis genes in controlling stem strength, we conducted a study measuring the stem bending strength (SBS) value of B73 and Mo17 at the 14th leaf stage. Our results showed that B73 had a higher SBS value compared to that of Mo17 (Figure 5A). Additionally, after examining the lignin staining of the two cross-sections, we observed that B73 had a higher degree of lignin staining than that of Mo17 (Figure 5B).

Furthermore, the gene expression patterns of these four gene combinations in the stems of B73 and Mo17 at the 14th leaf stage showed that three combinations showed higher expression levels in B73 than Mo17 except the *ZmTCPTF16/Zm00001eb098190* combination (Figure 5C). This suggests that there was a strong co-relationship between this gene expression and the higher lignin content in the stem. Interestingly, the combination of *ZmMYB19/Zm00001eb308860* was in the predicted MYB regulatory network.

## 3. Discussion

Lignin is a secondary metabolite that is widely distributed in various tissues of maize. Its biosynthesis and metabolism are part of the phenylpropanolysis pathway. This pathway involves a series of reactions that ultimately produce three lignin monomers (see Appendix A), which then undergo polymerization to form lignin. Approximately 10 types of enzymes are involved in regulating lignin synthesis. In addition to its role in lignin production, lignin also plays a crucial role in plant growth and development, as well as defense against biological and abiotic stressors [43,44]. Some progress has been made in the study of lignin synthase in maize. For instance, the genes *bm1* (CAD) [34], *bm3* (COMT) [45], *bm5* (4CL1) [32], *CCoAOMT2*, and *HCT* [46] have been identified. However, the exact copy number, characteristics, and expression patterns of these genes related to maize lignin biosynthesis are not fully understood. To reveal the genes involved in the maize lignin biosynthesis pathway, we used Arabidopsis genes as references to search lignin homologous genes in the maize genome and identified a total of 117 genes from 10 gene families involved in the maize lignin biosynthesis pathway in this study, which are distributed across different chromosomes. Evolutionary tree analysis revealed a high degree of homology between maize and Arabidopsis lignin synthesis proteins (Appendix A). The predicted PAL and CAD family genes in this study show significant overlap with those studied by others [45] and CAD family genes [46], affirming the reliability of the predicted genes in this study. It is a valuable method to genome-widely identify genes. Several genes in plants have been identified based on this method, such as flavonoid biosynthesis genes in Ginkgo biloba [47], Helix-Loop-Helix (bHLH) transcription factor family [48], GPAT gene family [49], and Poly-galacturonase gene family in maize [50], several valuable genes have been identified and for further dissecting the biological functions. So, this method could be used in gene identification for other crops, while the actual gene function needs more experiments to be confirmed, like gene editing [51,52,53], over-expression analysis [54], and so on [55].

Maize is a diploid crop with a genome size of about 2.3 Gb [56], but a large number of cytological, genetic, and genomic studies have proved that maize has undergone polyploid origin in evolutionary history, so the genes in the genome have multiple copies and complex functions [57]. The results of the gene copy number of lignin synthesis genes identified in this study confirmed this idea. Different gene copy numbers were identified for the lignin synthesis gene, such as the CCR gene family; this suggests a significant increase in the copy number of maize lignin biosynthesis genes. These genes usually have similar sequences or functions, which may be due to repeated replication or transposition events during their evolution. Therefore, this clustered distribution can lead to the formation of gene families, making them adjacent or close on chromosomes. The expression patterns of genes in different tissues can elucidate their varying levels of expression, enabling plants to execute diverse physiological processes. To elucidate the complete expression patterns of these identified synthetic genes, we analyzed their expression using RNA Seq data from 21 different tissues. Among these tissues, we found that the gene with the highest expression was in the root, followed by the seed; they are fundamental for maize growth and reproduction. In roots, lignin imparts hardness and compressive strength to the cell wall, providing the structural and mechanical support necessary for root tissue to penetrate the soil. Lignin also forms the plant’s vascular system, which plays a crucial role in water and nutrient absorption from the soil and their transportation to aboveground parts. Additionally, lignin aids in root resistance against pathogenic microorganisms [58,59] and other biological stresses. In seeds, lignin contributes to the formation of a robust seed coat, enhancing seed resistance and protecting them from environmental damage and invasion by pathogenic microorganisms, fungi, or insects, thus extending their survival in the soil. During seed maturation and dormancy, lignin helps maintain internal water balance and stable nutrient storage, ensuring seed survival and growth potential. Studies have indicated that lignin content in seeds can influence germination and seed success rates [60]. Therefore, these gene expression data can serve as a basis for further candidate gene analysis.

Lignin synthesis genes are not only regulated by their own synthetic genes but also by transcription factors (TFs), which are key regulatory factors for gene expression and have diverse functions in plants. TFs help plants respond to abiotic stress by modulating gene expression at the transcriptional level. Different types of TFs have been found to regulate lignin synthesis in various crops, impacting different biological events. Among these, MYB and NAC are particularly important in lignin synthesis regulation [61,62,63], altering their expression to modulate the lignin synthesis process. For instance, *ZmMYB31* and *ZmMYB42* are involved in regulating phenylpropanoid genes in maize, functioning at all stages of this pathway and in various tissues during seedling leaf development. In mature leaf tissues of maize, *Zm4CL2*, *ZmF5H*, and *ZmCOMT1* are common targets of *ZmMYB31* and *ZmMYB42*. The differential regulatory effects of these genes help redirect biological intermediates toward lignin production [64].

In this study, two methods were employed to predict the interaction between 151 transcription factors and 96 synthetic genes (Appendix A), and their combined accuracy surpassed that of a single method. Subsequent gene expression testing confirmed that potential transcription factors and synthetic gene pairs are crucial, warranting further investigation. The predicted results indicate that different MYBs or NACs regulate different types or quantities of synthetic genes. These findings suggest that, similar to many biological processes, lignin biosynthesis is a complex process regulated by various genetic factors with different regulatory mechanisms.

The content of lignin can be used as the main indicator to evaluate the lodging resistance of crops, and its content and related enzyme activity are significantly correlated with the lodging resistance of stems [65,66]. Therefore, the high expression of lignin synthesis genes is also positively correlated with plant lodging resistance. At the same time, some transcription factors are also involved in lignin synthesis and anti-lodging effects [67,68]. Due to the complex network regulation between transcription factors and synthetic genes, the genes with the highest expression of synthase genes and TFs in the stem were selected as candidate genes to explore the gene network of stem-specific regulation. Finally, four combinations were obtained, and their expression patterns were further verified by RT qPCR. Perhaps due to the incomplete morphology of the stem in the seedling stage, the staining situation in B73 and Mo17 stems was basically similar (Appendix A). These genes were not highly expressed in the stem (Appendix A), but most of the genes in B73 in the 14th leaf stage were highly expressed in the stem (Figure 4C), which is consistent with RNA. The Seq data are similar, indicating that genes with high stem-specific expression levels need to be validated at specific stages to ensure the accuracy of gene screening.

Due to the necessity of lignin polymerization during the development of vascular bundle thick-walled cells [30,69], this study found a positive correlation among the 14 leaf stage B73 and Mo17 stem strength measurements, lignin staining, and candidate gene expression detection, suggesting that high expression of lignin synthesis genes and transcription factors in stem increased lignin content and stem strength. The three identified TF pairs of lignin synthesis genes can serve as the focus of further research. Meanwhile, the research method of this study can be further applied to other crops.

## 4. Materials and Methods

### 4.1. Plant Materials

Two maize inbred lines, B73 and Mo17, were selected as materials to plant in the field trial experimental locations of the University of Science and Technology Beijing, Beijing, China, in 2023. The average stem bending strength values of the two inbred lines were collected during the milk ripening period. Collect the roots, stems, and leaves of the 6th leaf stage of both B73 and Mo17 maize plants, as well as the roots, stems, and leaves of the 14th leaf stage of B73, and the stem tissues of the 14th leaf stage of Mo17. Immediately freeze the samples in liquid nitrogen and store them at −80 °C to preserve RNA integrity. For the 6th leaf stage samples, separately combine the roots, stems, and leaves. For the 14th leaf stage samples, combine the main roots and fibrous roots, and for the stem tissues, combine the 3rd and 4th aboveground sections of the stem, as well as the top three slices. Subsequently, extract RNA from each tissue sample to determine the gene expression patterns. Each tissue sample should consist of a mix of three individual plants and the RNA extraction and gene expression analysis should be repeated three times to ensure accuracy and reproducibility of the results.

### 4.2. Identification of Gene Family Members and Sequence Analysis

The sequences of the synthetic pathway genes, *AtPAL*, *AtC4H*, *At4CL*, *AtHCT*, *AtC3H*, *AtCCoAMT*, *AtCCR*, *AtF5H*, *AtCOMT*, and *AtCAD* in *A. thaliana* were downloaded from the TAIR database (http://www.arabidopsis.org (accessed on 5 July 2023)) (Appendix A). Using the protein sequences of these genes as queries, the BLAST program (E-value < 1 × 10^−50^) in the Phytozome V13 database (https://phytozome-next.jgi.doe.gov/blast-search/ (accessed on 13 July 2023)) identified homologous genes in the maize genome. By integrating the analyzed results from these two methods, the candidate biosynthesis genes in maize were retrieved. The chromosomal localizations of all the identified gene family members were extracted from the MaizeGDB database and visualized using MapChart (v2.32).

Protein sequences were aligned using ClustalW in MEGA5.04 with default parameters for all lignin synthetic genes. Phylogenetic analysis of these genes was performed using the Neighbor-Joining (NJ) method with a bootstrap value of 1000 in MEGA5.04 (https://www.megasoftware.net/).

### 4.3. Potential Transcriptional Regulation Analysis of Lignin Biosynthesis Genes

The expression data of lignin synthesis genes in 21 specific tissues were retrieved from the MaizeGDB database (https://qteller.maizegdb.org/ (accessed on 28 July 2023)) (Appendix A). The tested tissues included Internode, vegetative meristem, ear primordium, embryo, endosperm, germinating kernels, pericarp aleurone, leaf zone, mature Leaf, 3 and 7 day old whole root system, mature pollen, female spikelet silk, and silk. The heatmap of gene expression is generated using the pheatmap package in the R Programming Language (R version 4.1.0).

To explore the potential transcriptional regulation pattern of these biosynthesis genes, the PlantTFDB database (http://planttfdb.gao-lab.org/ (accessed on 27 July 2023)) was first used to predict TFs related to the biosynthesis genes. The Triple Gene Mutual Interaction (TGMI) algorithm [70] was used to further analyze the possible interaction relationships based on the identified TFs and the synthetic genes by gene expression data retrieved from the MaizeGDB database.

### 4.4. Expression Pattern Analysis of Lignin Synthesis Gene and Visualization

RNAs from roots, stems, and leaves were extracted by using TransZol (TransGen Biotech, Beijing, China). Total RNA was reverse transcribed into template cDNA using All-in-one RT SuperMix kit (Vazyme, Nanjing, China). Based on the results of gene expression values in B73, 4 combinations with 8 genes, including *Zm00001eb099470*(*ZmBHLH88*), *Zm00001eb187810*(*ZmMYB19*), *Zm00001eb248270*(*ZmTCPTF16*), *Zm00001eb150550*(*ZmUMC2381*), *Zm00001eb186280*(*ZmWRKY98*), *Zm00001eb098190*, *Zm00001eb234090*, and *Zm00001eb308860* with high expression values in stem were selected for performing RT-qPCR analysis in the two represented inbred lines (Appendix A).

Gene-specific primers were designed using qprimerDB software (https://qprimerdb.biodb.org/analysis/ (accessed on 9 September 2023)). RT-qPCR assays were performed on QuantStudio 5 system (ABI, Natick, MA, USA) using SYBR TB GreenTM Premix (TaKaRa, Dalian, China) with *ZmActin* as an internal reference. Three biological replicates were set up for each experiment, and relative expression was calculated by 2^−ΔΔCt^ method [71].

### 4.5. Different Varieties Stained for Lignin in Stem Tissues

In order to compare the lignin content of the stems in B73 and Mo17, stem cross-section tissues from the 6th leaf and the 14th leaf stage of B73, as well as Mo17 14th leaf stages, were immersed in phloroglucinol staining solution (2.5 g phloroglucinol dissolved in 100 mL 95% ethanol) for 3 min. After removal, the surfaces were rinsed with ddH_2_O, followed by the addition of concentrated hydrochloric acid onto the stem cross-sections. After standing for 1 min, use Olympus SZX2-ILLB stereoscope (Olympus, Tokyo, Japan) to observe lignin staining in vascular bundles.

## 5. Conclusions

In the current study, a total of 117 synthesis-related genes of 10 gene families involved in the lignin biosynthesis pathway of the maize genome were identified. The copy numbers, distribution locations, and gene expression patterns were analyzed. Additionally, 151 TFs were predicted to interact with 96 synthesis genes through two methods. Of these TFs, 29 were members of two main types and potentially regulate 43 synthesis genes. To investigate the relationship between these identified genes and lodging resistance, four pairs of gene/TF combinations were screened based on their high expression values in the stem. RT-qPCR experiments showed that three combinations showed higher expression in the B73 stem compared to that in Mo17. The degree of lignin staining was also higher in B73, indicating that these genes may play a role in regulating lignin synthesis and have a potential anti-lodging effect. Additional studies, such as on gene knockout or overexpression experiments, are necessary to further investigate this hypothesis.

## Figures and Tables

**Figure 1 ijms-25-06710-f001:**
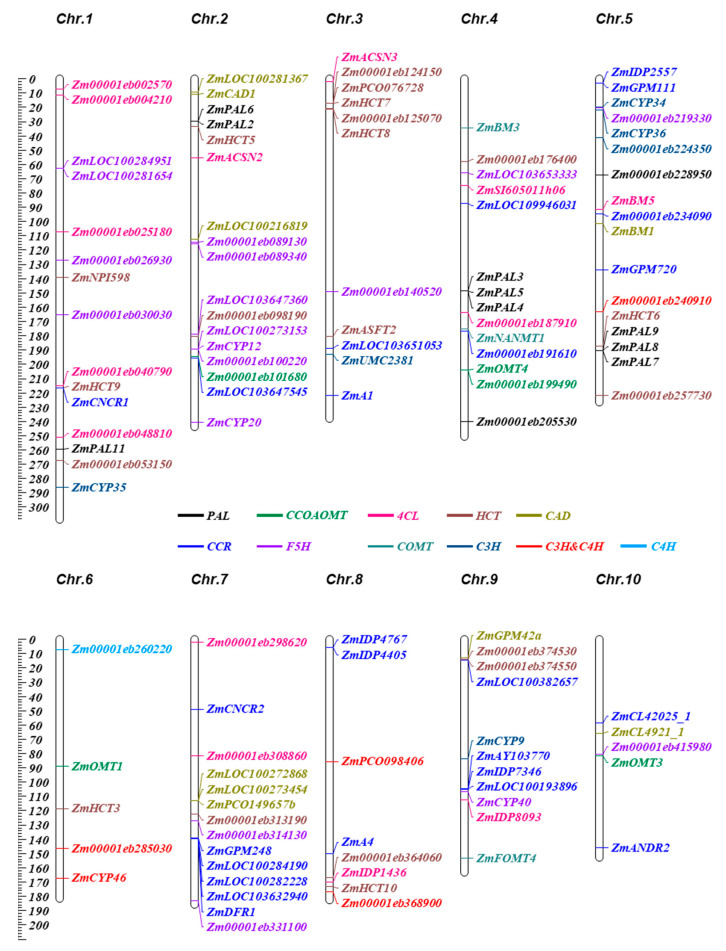
Distribution of lignin synthesis pathway-related genes in the maize genome. The chromosome number is displayed on the top of each chromosome, and the genetic marker is located on the right side. The different colors represent ten lignin synthesis genes. The leftmost scale indicates the chromosome length in Megabases (Mb).

**Figure 2 ijms-25-06710-f002:**
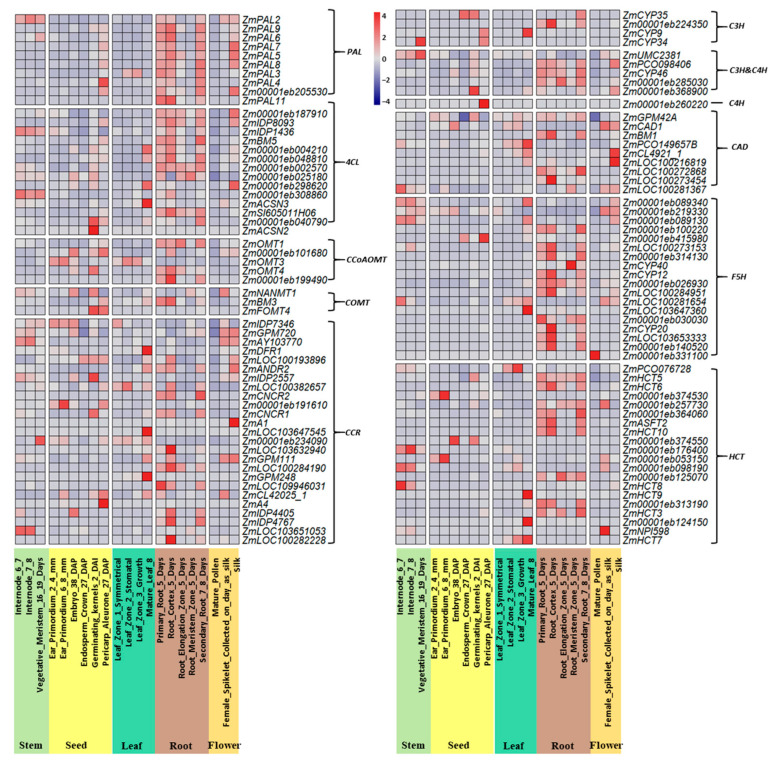
Lignin synthesis gene expression patterns in various tissue types. The bar on the side of the heatmap represents the log2 converted values.

**Figure 3 ijms-25-06710-f003:**
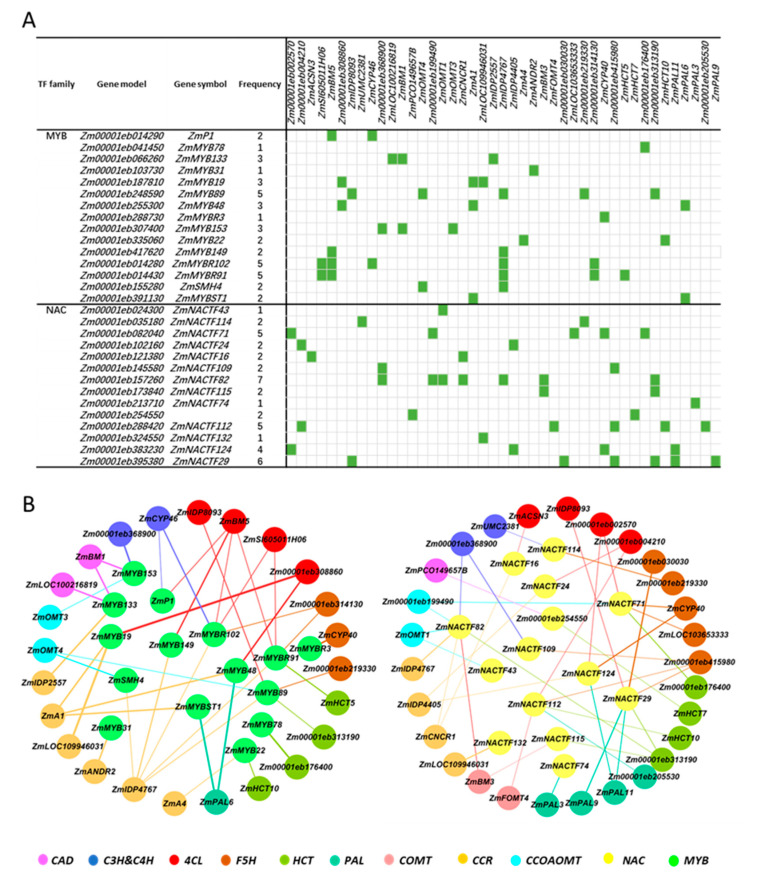
The transcriptional regulatory network between lignin biosynthesis genes and TFs using RNA-Seq data. (**A**): The regulatory relationship between the two main types of TFs and all corresponding lignin biosynthesis genes. The green blocks represent interactions with statistical significance. (**B**): The regulatory network of two main types of TFs and corresponding specific single lignin biosynthesis genes.

**Figure 4 ijms-25-06710-f004:**
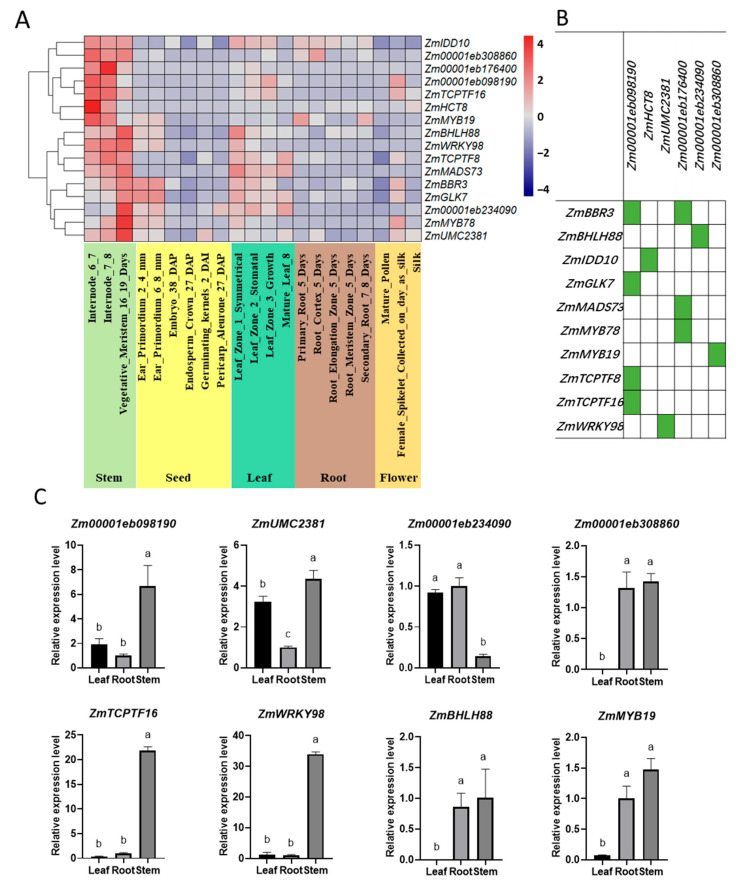
Screening of genes related to lignin synthesis pathway in stem. (**A**): Expression heatmaps of genes related to lignin biosynthesis pathway in different tissue types (Using the Euclidean distance measurement). (**B**): The regulatory relationship between TF and all corresponding lignin synthesis genes. (**C**): Gene expression patterns of lignin candidate genes. Data are presented as means ± SE (n = 3). Different letters above line graphs show a significant difference (*p* ≤ 0.05) among the two hybrids.

**Figure 5 ijms-25-06710-f005:**
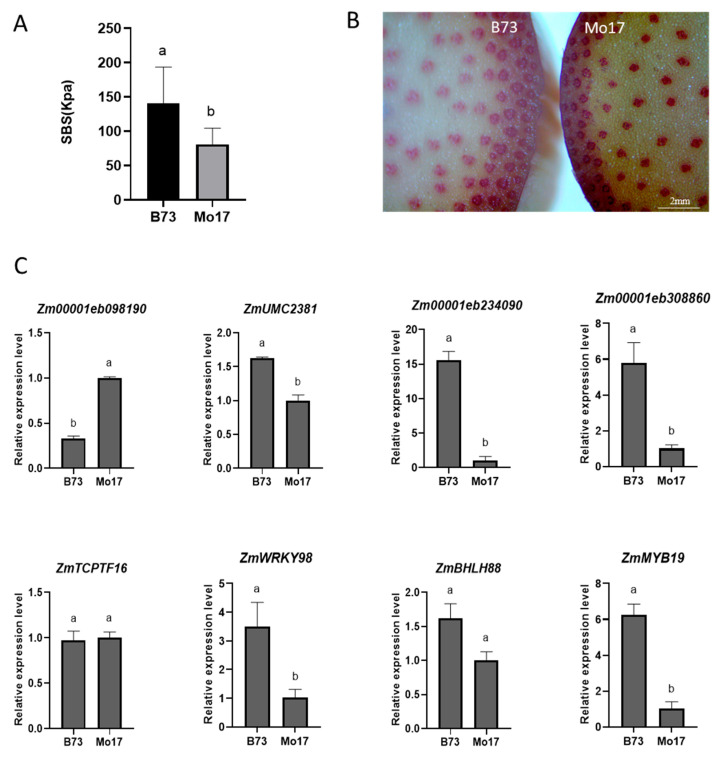
Lodging-related trait phenotype and differential gene analysis between B73 and Mo17. (**A**): Comparison of stem strength differences. N = 15. (**B**): Lignin staining of stems (Red range). (**C**): lignin synthesis gene expression in stems. Data are presented as means ± SE (n = 3). Different letters above line graphs show a significant difference (*p* ≤ 0.05) among the two hybrids.

## Data Availability

Data are contained within the article.

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
