# Peer review of "Genome-Wide Identification and Characterization of Lignin Synthesis Genes in Maize"

_ijms, 2024, doi:10.3390/ijms25126710_

Round 1

Reviewer 1 Report

Comments and Suggestions for Authors

The manuscript “Genome wide identification and characterization of lignin synthesis genes in maize” submitted by Wang et al., was carefully reviewed. Lignin is one of the main components of plant cell wall and plays a vital role in many aspects of plant growth. Lignin biosynthesis extensively contributes to plant growth, tissue/organ development, lodging resistance and the responses to a variety of biotic and abiotic stresses. In this paper, the number of copies, characteristics, and expression patterns of genes involved in lignin biosynthesis in maize was identified and analysis, which has specific reference value for understanding lignin related genes and their biological functions in herbaceous plants.

My concerns are as follows:

(1)      Some sentences in the abstract are too vague to know what they mean? Such as “The promoters of these genes contain various transcription factor binding elements.”, What are the key components? How many? Moreover, “and their interactions with transcription factors.” for what?

(2)      The Figure 1 should be deleted. Figure 1 is the results of previous studies, rather than the main content of this article. It is enough to cite this content. In addition, figure 1 also has no title.

(3)      For Figure3, the expression levels in different tissues vary greatly. Why is the expression level of lignin related genes very low in stems? It is higher in the root. How can this be explained by the expected goal.

(4)      For L152-155, What does “and silks,” mean? I didn't find the corresponding sample abbreviation in Figure 3A. It may be the “Flowers”?

(5)      L177-187, Please check the full text. Genes need to be italicized, such as "Zm00001eb248590, Zm00001eb014280, Zm00001eb014430" "Zm00001eb041450, Zm00001eb103730, and Zm00001eb288730", etc.

(6)      Figure 5 and Figure 6, the results of significant difference analysis are missing in picture c. The same in Supplementary Figure S2.

(7)      Promoter elements mentioned in the abstract? Why didn't find it in the results?

(8)      Maize genetic transformation is relatively easy, and the key genes screened in the results lack functional verification.

Author Response

The manuscript “Genome wide identification and characterization of lignin synthesis genes in maize” submitted by Wang et al., was carefully reviewed. Lignin is one of the main components of plant cell wall and plays a vital role in many aspects of plant growth. Lignin biosynthesis extensively contributes to plant growth, tissue/organ development, lodging resistance and the responses to a variety of biotic and abiotic stresses. In this paper, the number of copies, characteristics, and expression patterns of genes involved in lignin biosynthesis in maize was identified and analysis, which has specific reference value for understanding lignin related genes and their biological functions in herbaceous plants.

My concerns are as follows:

(1)Some sentences in the abstract are too vague to know what they mean? Such as “The promoters of these genes contain various transcription factor binding elements.”, What are the key components? How many? Moreover, “and their interactions with transcription factors.” for what?

Response: Thanks for your kind suggestion. We have carefully checked the abstract and deleted some sentences with vague meanings and added some detailed information, please see the abstract part.

(2)The Figure 1 should be deleted. Figure 1 is the results of previous studies, rather than the main content of this article. It is enough to cite this content. In addition, figure 1 also has no title.

Response: Thanks for your kind suggestion. According this suggestion, we have moved the Figure 1 to supplementary Figure 1. Please see the supplementary files.

(3)For Figure3, the expression levels in different tissues vary greatly. Why is the expression level of lignin related genes very low in stems? It is higher in the root. How can this be explained by the expected goal.

Response: Thanks for your kind suggestion. There were 5 root and 3 stem tissues included in the different tissues, so the number of genes with high expression values in roots were more than that in stems. After analyzing the gene expression values, we chose the genes with high expression values in stems to do further experiments to confirm.   

(4)For L152-155, What does “and silks,” mean? I didn't find the corresponding sample abbreviation in Figure 3A. It may be the “Flowers”?

Response: Thanks for your kind suggestion. I’m so sorry for this mistake, and we have revised the character “silk” to “flower”. Please see the content in L145-147. 

(5)L177-187, Please check the full text. Genes need to be italicized, such as "Zm00001eb248590, Zm00001eb014280, Zm00001eb014430" "Zm00001eb041450, Zm00001eb103730, and Zm00001eb288730", etc.

Response: Thanks for your kind suggestion. We have revised the writing styles as this suggestion. Please see the content in L169-171.

(6)Figure 5 and Figure 6, the results of significant difference analysis are missing in picture c. The same in Supplementary Figure S2.

Response: Thanks for your kind suggestion. We have added the significant difference analysis results in the current Figure 4(previous Figure 5), Figure5(previous Figure 6) and Supplementary Figure S3(previous Figure S2).  

(7)Promoter elements mentioned in the abstract? Why didn't find it in the results?

Response: Thanks for your kind suggestion. I’m sorry for this mistake and we have carefully checked the abstract and deleted this sentence.

(8)Maize genetic transformation is relatively easy, and the key genes screened in the results lack functional verification.

Response: Thanks for your kind suggestion. Actually, one of the aims of this ms is to discover the lignin synthetic genes in genome-wide level, and we will do maize genetic transformation later to confirm the candidate genes’ function. 

Reviewer 2 Report

Comments and Suggestions for Authors

Shuai Wang et al identified genes underlying lignin synthesis in maize using comparative genomics and rt-qPCR. The work is interesting, although the writing style is confusing and doesn't help to get a clear picture of the work's aims, results, and importance. 
- For example, lines 44-47 present results of previous work on rice and wheat. Why not use the synteny between maize and rice to help identify genes?
- The introduction needs to be rewritten and restructured to have a logical flow. For example, it should give an overall look at the pathway and then specific genes. Commonly, abbreviations should be used once introduced earlier. 
- Figure 1 is not mentioned in the text; I suddenly found it at the end of the introduction. 
The discussion is very short and general and doesn't really discuss and show the importance of the findings; it simply reports findings. 

Author Response

Shuai Wang et al identified genes underlying lignin synthesis in maize using comparative genomics and rt-qPCR. The work is interesting, although the writing style is confusing and doesn't help to get a clear picture of the work's aims, results, and importance. 

(1)For example, lines 44-47 present results of previous work on rice and wheat. Why not use the synteny between maize and rice to help identify genes?

  Response: Thanks for your kind suggestion and we think it’s really a good idea. We analyzed the lignin synthetic genes in rice and wheat genome and found there were fewer researchers in these two crops compared with that in Arabidopsis, so we selected Arabidopsis genes as synteny genes to do blast analysis. We think we could compare the lignin synthetic genes among these three cereal crops later to find the conserved genes.

(2)The introduction needs to be rewritten and restructured to have a logical flow. For example, it should give an overall look at the pathway and then specific genes. Commonly, abbreviations should be used once introduced earlier. 

Response: Thanks for your kind suggestion. We have rewritten and reconstructed the introduction part. Now the logic flows are as follows: 1)The importance and biological function of lignin; 2) The synthetic pathway and the involving genes; 3) Examples of specific synthetic genes; 4) The function of transcriptional factors involving in lignin synthesis;5) The research aim, content and meanings of this ms.

(3)Figure 1 is not mentioned in the text; I suddenly found it at the end of the introduction. 

Response: Thanks for your kind suggestion. We have moved the Figure 1 to supplementary Figure 1. Please see the supplementary files.
(4) The discussion is very short and general and doesn't really discuss and show the importance of the findings; it simply reports findings. 
  Response:  Thanks for your kind suggestion. According to this suggestion, we rewritten the discussion part. Now the logic flows are as follows: 1) The reliability of the gene identifying methods used in this study; 2) Why the number of genes with high expression values in roots were relatively higher than that in stems; 3) The reliability and advantages of the gene and TF interaction identifying methods used in this study; 4) the candidate lignin genes identified in this study could provide further investigation clues.

Reviewer 3 Report

Comments and Suggestions for Authors

Dear Authors,

in your study you used "two representative maize imbred lines". Based on which parameters did you assess their representativeness? What signs, properties are they different? I am asking for a closer characterization of the lines used in the M&M section. Please explain why the samples of different tissues were taken in different developmental stages only for one line and for the other only in one stage? Characterize the samples? Was it a pooled sample? How many samples were taken? Were the plants chosen randomly, or were selection criteria selected?

Please, specify the BLAST algorithm used, since the authors performed the alignment at the amino acid level. It is also necessary to provide a link.

Please, provide the link for the heatmap construction (line 298).

Please, specify the background of the RNA extraction. Were samples from only one line used for RNA isolation? List the genes which were analyzed.

Please, include the reference for the for microscopic methodology (line 314) and specify the microscope and the scale of observation.

Author Response

(1)In your study you used "two representative maize inbred lines". Based on which parameters did you assess their representativeness? What signs, properties are they different? I am asking for a closer characterization of the lines used in the M&M section.

Response: Thanks for your this suggestion. The two inbred lines, B73 and Mo17, are two representative maize inbred lines used in maize genetic and genomics researches. Because these two lines are the two lines having high-quality genomic data, meanwhile different genes controlling agronomic traits have been identified based on these two lines.

(2)Please explain why the samples of different tissues were taken in different developmental stages only for one line and for the other only in one stage? Characterize the samples? Was it a pooled sample? How many samples were taken? Were the plants chosen randomly, or were selection criteria selected?

Response: Thanks for your kind suggestion. Based on the suggestion and question, we added the corresponding words in the text to show how the samples were extracted, and how the RNA was extraction in the M&M part. Please see the words in L335-L340.

(3)Please, specify the BLAST algorithm used, since the authors performed the alignment at the amino acid level. It is also necessary to provide a link.

Response: Thanks for your kind suggestion. We have added the corresponding link in L345-346.

(4)Please, provide the link for the heatmap construction (line 298).

Response: Thanks for your kind suggestion. We have added the link for the heatmap construction in 360-361,and the data was listed in Supplementary Table S3.

(5)Please, specify the background of the RNA extraction. Were samples from only one line used for RNA isolation? List the genes which were analyzed.

Response: Thanks for your kind suggestion. We have added the background of the RNA extraction. For RNA extraction, the stem tissues in the V14 stages were used. And the information was added in Supplementary table5, also the words were added in L371-375.

(6)Please, include the reference for the for microscopic methodology (line 314) and specify the microscope and the scale of observation.

  Response: Thanks for your kind suggestion. We have added the corresponding methods and references in L388-390

Reviewer 4 Report

Comments and Suggestions for Authors

The manuscript reported the bioinformatic analysis and the gene expression analysis to discover the lignin synthetic genes. Three potential stem lodging resistance genes and their interactions with transcription factors were identified. 

General concept comments

The manuscript shows typing errors, gaps, or missing points. The experimental design is appropriate. Materials and methods are not completely described, therefore the results based on the details given in the methods section could be not totally reproducible. The discussion should be improved. The cited references are not recent publications. The conclusions are missing, data availability statements were not reported. According to these considerations I suggest a major revision.

Specific comments

Materials and methods: The exact sampling location of the plant material shall be reported. Moreover, how was the obtained plant material used?

Figures: 

Line 107: figure 1 caption should be moved below the figure. Moreover, figure 1 reported some names with the lowercase letter.

Figure 2 seems to report specific position on the chromosomes named “Chr.” not mentioned earlier in the text.

Some of the figures are not visible. I suggest replacing them with better solved ones. More specifically, figures 3-4-5-6c data are not visible and not comprehensive. 

The discussion should be improved. The authors should discuss the results and how they can be interpreted in perspective of previous studies and of the working hypotheses. The limitations of the work shouldbe highlighted and future research directions may also be mentioned. 

The conclusions are missing. I suggest adding this section.

Comments on the Quality of English Language

Moderate editing of English language required

Author Response

The manuscript reported the bioinformatic analysis and the gene expression analysis to discover the lignin synthetic genes. Three potential stem lodging resistance genes and their interactions with transcription factors were identified. 

General concept comments

The manuscript shows typing errors, gaps, or missing points. The experimental design is appropriate. Materials and methods are not completely described, therefore the results based on the details given in the methods section could be not totally reproducible. The discussion should be improved. The cited references are not recent publications. The conclusions are missing, data availability statements were not reported. According to these considerations I suggest a major revision.

Specific comments

(1)Materials and methods: The exact sampling location of the plant material shall be reported. Moreover, how was the obtained plant material used?

Figures: 

  Response: Thanks for your kind suggestion. We have carefully checked the M&M part and revised the writing, please see the content in L326-340

(2)Line 107: figure 1 caption should be moved below the figure. Moreover, figure 1 reported some names with the lowercase letter.

  Response: Thanks for your kind suggestion. According this suggestion, we have moved the Figure 1 to supplementary Figure 1. Please see the supplementary files.

(3)Figure 2 seems to report specific position on the chromosomes named “Chr.” not mentioned earlier in the text.

   Response: Thanks for your kind suggestion. We have added the abbreviation of Chr. to chromosomes in L116.

(4)Some of the figures are not visible. I suggest replacing them with better solved ones. More specifically, figures 3-4-5-6c data are not visible and not comprehensive. 

   Response: Thanks for your kind suggestion. We have adjusted the resolution for these three figures and inserted them in the text.

(5)The discussion should be improved. The authors should discuss the results and how they can be interpreted in perspective of previous studies and of the working hypotheses. The limitations of the work should be highlighted and future research directions may also be mentioned. 

  Response: Thanks for your kind suggestion. According to this suggestion, we rewritten the discussion part. Now the logic flows are as follows: 1) The reliability of the gene identifying methods used in this study; 2) Why the number of genes with high expression values in roots were relatively higher than that in stems; 3) The reliability and advantages of the gene and TF interaction identifying methods used in this study; 4) the candidate lignin genes identified in this study could provide further investigation clues.

(6)The conclusions are missing. I suggest adding this section.

  Response: Thanks for your kind suggestion. We have added the conclusion part in the ms, please see the content in L214-L226.

Round 2

Reviewer 1 Report

Comments and Suggestions for Authors

This version has answered all my questions and the current manuscript is suitable for publication.

Author Response

Thank you for your recognition of my research.

Reviewer 2 Report

Comments and Suggestions for Authors

The manuscript reads much better now.

Author Response

(The authors gave the same response as above.)

Reviewer 4 Report

Comments and Suggestions for Authors

The manuscript has been improved. However, the conclusion section should be moved below the materials and methods section.

Author Response

Comments and Suggestions for Authors

The manuscript has been improved. However, the conclusion section should be moved below the materials and methods section.

Response: Thanks for your kind suggestion. We have adjusted the position of the conclusion.